# Design, synthesis and biological evaluation of *N*-oxide derivatives with potent *in vivo* antileishmanial activity

**Leandro da Costa Clementino**[1,2], **Guilherme Felipe Santos Fernandes**[1,2], **Igor Muccilo Prokopczyk**[2], **Wilquer Castro Laurindo**[1,2], **Danyelle Toyama**[3], **Bruno Pereira Motta**[2], **Amanda Martins Baviera**[2], **Flávio Henrique-Silva**[3], **Jean Leandro dos Santos**[2]*, **Marcia A. S. Graminha**[2]*

1 São Paulo State University (UNESP), Institute of Chemistry, Araraquara, Brazil, 2 São Paulo State University (UNESP), School of Pharmaceutical Sciences, Araraquara, Brazil, 3 Department of Genetics and Evolution, Federal University of São Carlos, São Carlos, Brazil

* jean.santos@unesp.br (JLS); marcia.graminha@unesp.br (MASG)

**Data Availability Statement:** All relevant data are within the paper and its Supporting Information files.

## Abstract

Leishmaniasis is a neglected disease that affects 12 million people living mainly in developing countries. Herein, 24 new *N*-oxide-containing compounds were synthesized followed by *in vitro* and *in vivo* evaluation of their antileishmanial activity. Compound **4f**, a furoxan derivative, was particularly remarkable in this regard, with $EC_{50}$ value of 3.6 μM against *L. infantum* amastigote forms and $CC_{50}$ value superior to 500 μM against murine peritoneal macrophages. *In vitro* studies suggested that **4f** may act by a dual effect, by releasing nitric oxide after biotransformation and by inhibiting cysteine protease CPB ($IC_{50}$: 4.5 μM). *In vivo* studies using an acute model of infection showed that compound **4f** at 7.7 mg/Kg reduced ~90% of parasite burden in the liver and spleen of *L. infantum*-infected BALB/c mice. Altogether, these outcomes highlight furoxan **4f** as a promising compound for further evaluation as an antileishmanial agent.

## Introduction

Leishmaniasis is a neglected disease caused by more than 20 species of the genus *Leishmania* and affects 12 million individuals living mainly in developing countries [1–3]. The current treatment still relies on a small number of old and toxic drugs, administered through the parenteral route, except for the oral drug miltefosine. The drug discovery pipeline for leishmaniasis has been affected by the severe scarcity of new drug candidates achieving clinical trials stages [4]. Therefore, efforts to discover new therapeutic alternatives are urgent.

Leishmaniasis is transmitted through the inoculation of promastigotes during the phlebotomines blood meal, followed by their entry and differentiation to amastigotes into mammalian hosts cells, primarily "professional phagocytes" such as macrophages [5]. The parasite has developed different strategies to inactivate the macrophages functions, including the expression and secretion of cysteine protease (CPB) [6]. CPB, as well as the *Trypanosoma cruzi*

**Funding:** We are grateful to The São Paulo Research Foundation, FAPESP, https://fapesp.br/en, (2017/03552-5, 2016/06931-4 and 2018/11079-0) and National Council for Scientific and Technological Development, CNPq, https://www.gov.br/cnpq/pt-br, (305174/2020-7, 304731-2017-0 and 311746/2017-9) for financial support. This study was financed in part by the Coordenação de Aperfeiçoamento de Pessoal de Nível Superior (CAPES), Brazil, under finance code 001. The funders had no role in study design, data collection and analysis, decision to publish, or preparation of the manuscript.

enzyme cruzain, is a cathepsin L-like protease important for parasite immune evasion [6, 7]. For *Leishmania*, CPB also allows modulation of the host Th1/Th2 immune response [8, 9] and degradation of MHC class I [10]. The importance of CPB for parasite infection and the lack of redundancy into the mammalian host genome, make it an attractive target for antileishmanial drug development [7, 11]. Several protease inhibitors exhibit electrophilic warheads that undergo nucleophilic attack from thiolate residue of cysteines of the active site of the enzymes [12–19]. Among them, the warhead *N*-acylhydrazone present protease inhibitory properties and has been previously explored in the synthesis of both antileishmanial and antitrypanocidal compounds [17, 18, 20]. Therefore, in a continuing effort to develop new candidate drugs to treat leishmaniasis, we have first reported a series of furoxan and benzofuroxan derivatives containing the *N*-acylhydrazone subunit as antileishmanial agents [21].

Among them, the compound **Lapdesf14e** (*E*)-4-(4-((2-(3-hydroxybenzoyl)hydrazono)methyl)phenoxy)-3-(phenylsulfonyl)-1,2,5-oxadiazole 2-oxide, previously known as **14e** (Fig 1), showed remarkable antileishmanial efficacy in *L. infantum*–infected hamsters by reducing the parasite load in the spleen (49.9%) and liver (54.2%), without toxic effects [14]. It was suggested that the antileishmanial activity of this furoxan derivative (**Lapdesf14e**) was related to its ability to release nitric oxide (NO) after biotransformation, which ultimately leads to parasite death mainly through DNA damage [22]. Further investigations have revealed that **Lapdesf14e** is also targeting the parasite enzyme CPB at micromolar range [20]. Therefore, based on these promising results, we designed a novel series of *N*-oxide compounds containing *N*-acylhydrazone subunit, represented by phenyl-furoxan (**4a-4g**), amide-furoxan (**4h-4o**), and benzofuroxan (**4p-4x**) based on **Lapdesf14e** (Fig 1).

Moreover, to evaluate the contribution of *N*-oxide subunit, an additional series of analogs containing only the *N*-acylhydrazone subunit (**14a-14g**) was also synthesized, excluding the furoxan and benzofuroxan moieties. Thus, based on the attempt to optimize **Lapdesf14e**, the present study reports the design, synthesis, and biological evaluation of a novel series of *N*-oxide derivatives (**4a-x**) and *N*-acylhydrazones (**14a-g**), followed by their antileishmanial and inhibitory effect against the recombinant *Leishmania* cysteine protease LmCPB2.8ΔCTE (CPB), expressed in the yeast *Pichia pastoris*.

## Materials and methods

### Chemistry

Reagents and solvents were purchased from commercial suppliers and used as received. Thin-layer chromatography (TLC), precoated with silica gel 60 (HF-254; Merck) to a thickness of 0.25 mm, was used for monitoring all reactions. The plates were revealed under UV light (254 nm) and, when necessary, treated with iodine vapor. All compounds were purified on a chromatography column with silica gel (60 Å pore size, 35-75-μM particle size) and the following solvents were used as mobile phase: dichloromethane, hexane, ethyl acetate, and petroleum ether. The purity was analyzed by HPLC using a Shimadzu LC-10AD chromatograph equipped with a model SPD-10A UV–vis detector (Shimadzu). The purity was superior to 98%. Melting points (mp) were determined in open capillary tubes using an electrothermal melting point apparatus (SMP3; Bibby Stuart Scientific). Infrared (IR) spectroscopy (KBr disc) was performed on an FTIR-8300 Shimadzu spectrometer, and the frequencies are expressed per cm$^{-1}$. The nuclear magnetic resonance (NMR) spectra for $^1$H and $^{13}$C of all compounds were obtained on a Bruker DRX-600 (600 MHz) NMR spectrometer using deuterated dimethyl sulfoxide (DMSO-d$_6$) or chloroform (CDCl$_3$) as a solvent for sample preparation. Chemical shifts were expressed in parts per million (ppm) relative to tetramethylsilane. The coupling constants are reported in hertz (Hz) and the signal multiplicities are reported as

**Fig 1. Drug design of the new compounds.**

singlet (s), doublet (d), doublet of doublet (dd), doublet of doublet doublets (ddd), triplet (t), and multiplet (m). Elemental analyses (C, H, and N) were performed on a Perkin-Elmer model 240C analyzer. All data were expressed within ± 0.4% of the theoretical values. Compounds **3**, **9**, **12**, **14b**, and **14e** were prepared according to previously described methodologies [23–27]. Compounds **4p-x** were synthesized according to previously described methods and the characterization data are not shown here [24].

**General procedures for the synthesis of compounds (4a-o).** The coupling reaction for preparation of compounds (**4a**-**o**) involved the treatment of aldehyde group (**3**, **9**, **12**) (1.5 mmol) with previously selected hydrazides (**5a-e**; **6a-b**) (1.5 mmol) in 15 mL of ethanol, catalyzed by hydrochloric acid 37% (0.2 mL). The reactions were stirred at room temperature up to 24h and monitored by TLC until the reactive consumption. After, the solvent was partially removed under reduced pressure, followed by the addition of iced water (~ 15 mL) to precipitate the final *N*-acylhydrazones (**4a**-**o**). The precipitate was collected by filtration and washed with cold water to provide the compounds with yields ranging from 82–96%. When necessary, further purification was performed using column chromatography (flash silica, eluent: 50% ethyl acetate; 50% hexane).

*(E)-4-(4-((2-benzoylhydrazono)methyl)phenoxy)-3-phenyl-1,2,5-oxadiazole 2-oxide (4a).* Off-white solid; yield 91%; mp: 210–213˚C. $^1$H NMR (600 MHz, DMSO-$d_6$) δ: 11.94 (s, 1H), 8.49 (s, 1H), 8.09 (d, *J* = 7.3 Hz, 2H), 7.90 (dd, *J* = 31.5, 7.2 Hz, 4H), 7.63 (d, *J* = 7.5 Hz, 4H), 7.60 (d, *J* = 6.7 Hz, 2H), 7.54 (t, *J* = 6.8 Hz, 2H). $^{13}$C NMR (151 MHz, DMSO-$d_6$) δ: 163.2, 161.9, 153.8, 146.8, 133.3, 132.6, 131.8, 130.9, 129.1, 128.8, 128.5, 127.6, 126.7, 121.7, 120.6, 108.4. Anal. Calcd (%) for $C_{22}H_{16}N_4O_4$: C: 66.00; H: 4.03; N: 15.98. Found: C: 66.03; H: 4.02; N: 15.98.

*(E)-4-(4-((2-(4-hydroxybenzoyl)hydrazono)methyl)phenoxy)-3-phenyl-1,2,5-oxadiazole 2-oxide (4b).* Off-white solid; yield 90%; mp: 223–225˚C. $^1$H NMR (600 MHz, DMSO-$d_6$) δ: 11.73 (s, 1H), 10.16 (s, 1H), 8.46 (s, 1H), 8.08 (d, *J* = 7.4 Hz, 2H), 7.84 (d, *J* = 7.5 Hz, 2H), 7.81 (d, *J* = 8.5 Hz, 2H), 7.66–7.57 (m, 5H), 6.87 (d, *J* = 8.5 Hz, 2H). $^{13}$C NMR (151 MHz, DMSO-$d_6$) δ: 163.0, 162.2, 161.0, 153.9, 145.8, 133.0, 131.2, 130.0, 129.4, 128.9, 127.0, 124.0, 121.9, 120.8, 115.3, 108.6. Anal. Calcd (%) for $C_{22}H_{16}N_4O_5$: C: 63.46; H: 3.87; N: 19.21. Found: C: 63.46; H: 3.87; N: 19.23.

*(E)-4-(4-((2-(4-aminobenzoyl)hydrazono)methyl)phenoxy)-3-phenyl-1,2,5-oxadiazole 2-oxide (4c).* Off-white solid; yield 93%; mp: 219–222˚C. $^1$H NMR (600 MHz, DMSO-$d_6$) δ: 11.53 (s, 1H), 8.43 (s, 1H), 8.08 (d, *J* = 7.3 Hz, 2H), 7.82 (d, *J* = 8.0 Hz, 2H), 7.67 (d, *J* = 8.4 Hz,

2H), 7.65–7.57 (m, 5H), 6.59 (d, $J$ = 8.5 Hz, 2H), 5.81 (s, $J$ = 8.4 Hz, 2H). $^{13}$C NMR (151 MHz, DMSO-$d_6$) δ: 161.9, 153.4, 152,4, 144.6, 133,0, 131.7, 130.9, 129.4, 129.1, 128.4, 126.7, 121.7, 120.5, 119.3, 112.6, 108.4. Anal. Calcd (%) for $C_{22}H_{17}N_5O_4$: C: 63.61; H: 4.13; N: 15.41. Found: C: 63.60; H: 4.13; N: 15.41.

*(E)-4-(4-((2-(4-nitrobenzoyl)hydrazono)methyl)phenoxy)-3-phenyl-1,2,5-oxadiazole 2-oxide (4d)*. Off-white solid; yield 82%; mp: 226–228°C. $^1$H NMR (600 MHz, DMSO-$d_6$) δ: 12.23 (s, 1H), 8.51 (s, 1H), 8.39 (d, $J$ = 8.7 Hz, 2H), 8.16 (d, $J$ = 8.7 Hz, 2H), 8.08 (d, $J$ = 7.3 Hz, 2H), 7.90 (d, $J$ = 8.6 Hz, 2H), 7.66–7.59 (m, 5H). $^{13}$C NMR (151 MHz, DMSO-$d_6$) δ: 161.8., 154.0, 149.3, 147.7, 139.0, 132.3, 130.9, 129.2, 129.1, 129.0, 126.7, 123.7, 121.7, 120.6, 108.4. Anal. Calcd (%) for $C_{22}H_{15}N_5O_6$: C: 59.33; H: 3.39; N: 21.55. Found: C: 59.35; H: 3.41; N: 21.55.

*(E)-4-(4-((2-(4-(tert-butyl)benzoyl)hydrazono)methyl)phenoxy)-3-phenyl-1,2,5-oxadiazole 2-oxide (4e)*. Off-white solid; yield 95%; mp: 223–227°C. $^1$H NMR (600 MHz, DMSO-$d_6$) δ: 11.87 (s, 1H), 8.48 (s, 1H), 8.09 (d, $J$ = 7.5 Hz, 2H), 7.86 (d, $J$ = 8.0 Hz, 4H), 7.65–7.59 (m, 5H), 7.55 (d, $J$ = 8.0 Hz, 2H), 1.32 (s, 9H). $^{13}$C NMR (151 MHz, DMSO-$d_6$) δ: 163.2, 162.0, 154.8, 153.8, 146.4, 132.8, 131.1, 130.7, 129.3, 128.9, 127.7, 126.8, 126.8, 125.4, 121.8, 120.7, 108.5, 31.1. Anal. Calcd (%) for $C_{26}H_{24}N_4O_4$: C: 68.41; H: 5.30; N: 14.02. Found: C: 68.40; H: 5.32; N: 14.04.

*(E)-4-(4-((2-carbamoylhydrazono)methyl)phenoxy)-3-phenyl-1,2,5-oxadiazole 2-oxide (4f)*. Off-white solid; yield 96%; mp: 196–198°C. $^1$H NMR (600 MHz, DMSO-$d_6$) δ: 11.51 (s, 1H), 8.26 (s, 1H), 8.11–8.05 (m, 4H), 7.95 (d, $J$ = 8.6 Hz, 2H), 7.65–7.59 (m, 3H), 7.56 (d, $J$ = 8.7 Hz, 2H). $^{13}$C NMR (151 MHz, DMSO-$d_6$) δ: 161.9., 153.6, 140.9, 132.5, 130.9, 129.1, 129.0, 126.7, 121.7, 120.4, 108.4. Anal. Calcd (%) for $C_{16}H_{13}N_5O_4$: C: 56.64; H: 3.86; N: 18.86. Found: C: 56.64; H: 3.89; N: 18.86.

*(E)-4-(4-((2-carbamothioylhydrazono)methyl)phenoxy)-3-phenyl-1,2,5-oxadiazole 2-oxide (4g)*. Off-white solid; yield 89%; mp: 193–195°C. $^1$H NMR (600 MHz, DMSO-$d_6$) δ: 10.33 (s, 1H), 8.08 (d, $J$ = 7.3 Hz, 2H), 7.86 (d, $J$ = 8.6 Hz, 3H), 7.65–7.58 (m, 3H), 7.53 (d, $J$ = 8.6 Hz, 2H), 6.56 (s, 2H). $^{13}$C NMR (151 MHz, DMSO-$d_6$) δ: 162.0, 156.7, 152.9, 137.8, 133.1, 130.9, 129.1, 128.1, 126.7, 121.7, 120.3, 108.3. Anal. Calcd (%) for $C_{16}H_{13}N_5O_3S$: C: 54.08; H: 3.69; N: 13.51. Found: C: 54.11; H: 3.71; N: 13.51.

*(E)-4-((4-((2-benzoylhydrazono)methyl)phenoxy)methyl)-3-carbamoyl-1,2,5-oxadiazole 2-oxide (4h)*. Off-white solid; yield 93%; mp: 221–225°C. $^1$H NMR (600 MHz, DMSO-$d_6$) δ: 11.77 (s, 1H), 8.46 (d, $J$ = 59.2 Hz, 2H), 7.91 (d, $J$ = 6.4 Hz, 2H), 7.85 (s, 1H), 7.70 (d, $J$ = 7.4 Hz, 2H), 7.59 (d, $J$ = 6.1 Hz, 1H), 7.53 (d, $J$ = 6.1 Hz, 2H), 7.15 (d, $J$ = 7.4 Hz, 2H), 5.48 (s, 2H). $^{13}$C NMR (151 MHz, DMSO-$d_6$) δ: 163.0, 159.1, 155.7, 155.1, 147.4, 133.5, 131.7, 128.7, 128.5, 127.6, 115.2, 110.5, 61.2. Anal. Calcd (%) for $C_{18}H_{15}N_5O_5$: C: 56.69; H: 3.96; N: 20.98. Found: C: 56.70; H: 3.98; N: 20.99.

*(E)-3-carbamoyl-4-((4-((2-(4-hydroxybenzoyl)hydrazono)methyl)phenoxy)methyl)-1,2,5-oxadiazole 2-oxide (4i)*. Off-white solid; yield 90%; mp: 201–203°C. $^1$H NMR (600 MHz, DMSO-$d_6$) δ: 11.56 (s, 1H), 10.13 (s, 1H), 8.44 (d, $J$ = 76.8 Hz, 2H), 7.85 (s, 1H), 7.79 (d, $J$ = 8.1 Hz, 2H), 7.68 (d, $J$ = 7.1 Hz, 2H), 7.13 (d, $J$ = 7.8 Hz, 2H), 6.85 (d, $J$ = 7.8 Hz, 2H), 6.77 (d, $J$ = 8.1 Hz, 1H), 5.47 (s, 2H). $^{13}$C NMR (151 MHz, DMSO-$d_6$) δ: 162.6, 160.6, 160.0, 158.9, 146.5, 129.6, 128.8, 128.5, 128.1, 124.0, 115.2, 115.0, 110.5, 61.2. Anal. Calcd (%) for $C_{18}H_{15}N_5O_6$: C: 54.41; H: 3.81; N: 24.16. Found: C: 54.39; H: 3.81; N: 24.16.

*(E)-4-((4-((2-(4-aminobenzoyl)hydrazono)methyl)phenoxy)methyl)-3-carbamoyl-1,2,5-oxadiazole 2-oxide (4j)*. Off-white solid; yield 91%; mp: 209–214°C. $^1$H NMR (600 MHz, DMSO-$d_6$) δ: 11.36 (s, 1H), 8.51 (s, 1H), 8.35 (s, 1H), 7.85 (s, 1H), 7.65 (d, $J$ = 7.8 Hz, 4H), 7.13 (d, $J$ = 8.4 Hz, 2H), 6.58 (d, $J$ = 8.3 Hz, 2H), 5.77 (s, 2H), 5.47 (s, 2H). $^{13}$C NMR (151 MHz, DMSO-$d_6$) δ: 162.8, 158.7, 155.7, 155.2, 152.2, 145.6, 129.3, 128.4, 128.3, 119.6, 115.1, 112.6, 110.5, 61.2. Anal. Calcd (%) for $C_{18}H_{16}N_6O_5$: C: 54.55; H: 4.07; N: 20.18. Found: C: 54.56; H: 4.06; N: 20.19.

*(E)-3-carbamoyl-4-((4-((2-(4-nitrobenzoyl)hydrazono)methyl)phenoxy)methyl)-1,2,5-oxa-diazole 2-oxide (4l)*. Off-white solid; yield 84%; mp: 229–233˚C. [1]H NMR (600 MHz, DMSO-$d_6$) δ: 12.07 (s, 1H), 8.51 (s, 1H), 8.43 (s, 1H), 8.38 (d, *J* = 8.4 Hz, 2H), 8.15 (d, *J* = 8.4 Hz, 2H), 7.85 (s, 1H), 7.72 (d, *J* = 8.3 Hz, 2H), 7.16 (d, *J* = 8.3 Hz, 2H), 5.49 (s, 2H). [13]C NMR (151 MHz, DMSO-$d_6$) δ: 161.3, 159.3, 155.7, 148.6, 139.2, 131.8, 129.1, 128.9, 128.4, 123.7, 123.6, 115.2, 110.5, 61.2. Anal. Calcd (%) for $C_{18}H_{14}N_6O_7$: C: 50.71; H: 3.31; N: 26.27. Found: C: 50.73; H: 3.31; N: 26.28.

*(E)-4-((4-((2-(4-(tert-butyl)benzoyl)hydrazono)methyl)phenoxy)methyl)-3-carbamoyl-1,2,5-oxadiazole 2-oxide (4m)*. Off-white solid; yield 89%; mp: 189–192˚C. [1]H NMR (600 MHz, DMSO-$d_6$) δ: 11.70 (s, 1H), 8.51 (s, 1H), 8.40 (s, 1H), 7.84 (d, *J* = 8.2 Hz, 3H), 7.69 (d, *J* = 8.0 Hz, 2H), 7.54 (d, *J* = 7.8 Hz, 2H), 7.14 (d, *J* = 8.0 Hz, 2H), 5.48 (s, 2H), 1.31 (s, 9H). [13]C NMR (151 MHz, DMSO-$d_6$) δ: 162.9, 159.0, 155.7, 155.1, 154.5, 147.1, 130.7, 128.7, 127.9, 127.4, 125.3, 115.2, 110.5, 61.2, 34.7, 30.9. Anal. Calcd (%) for $C_{22}H_{23}N_5O_5$: C: 60.40; H: 5.30; N: 18.29. Found: C: 60.40; H: 5.33; N: 18.31.

*(E)-3-carbamoyl-4-((4-((2-carbamoylhydrazono)methyl)phenoxy)methyl)-1,2,5-oxadiazole 2-oxide (4n)*. Off-white solid; yield 96%; mp: 228–230˚C. [1]H NMR (600 MHz, DMSO-$d_6$) δ: 11.36 (s, 1H), 8.50 (s, 1H), 8.14 (s, 1H), 7.98 (d, *J* = 24.2 Hz, 2H), 7.85 (s, 1H), 7.76 (d, *J* = 8.3 Hz, 2H), 7.07 (d, *J* = 8.3 Hz, 2H), 5.46 (s, 2H). [13]C NMR (151 MHz, DMSO-$d_6$) δ: 158.9, 155.7, 155.1, 141.9, 131.8, 128.9, 127.8, 115.0, 110.5, 61.2. Anal. Calcd (%) for $C_{12}H_{12}N_6O_5$: C: 45.00; H: 3.78; N: 24.98. Found: C: 45.03; H: 3.80; N: 24.99.

*(E)-4-((4-((2-carbamothioylhydrazono)methyl)phenoxy)methyl)-3-carbamoyl-1,2,5-oxadia-zole 2-oxide (4o)*. Off-white solid; yield 90%; mp: 232–234˚C. [1]H NMR (600 MHz, DMSO-$d_6$) δ: 10.15 (s, 1H), 8.50 (s, 1H), 7.82 (d, *J* = 37.5 Hz, 2H), 7.67 (d, *J* = 8.4 Hz, 2H), 7.05 (d, *J* = 8.4 Hz, 2H), 6.45 (s, 2H), 5.44 (s, 2H). [13]C NMR (151 MHz, DMSO-$d_6$) δ: 158.2, 156.8, 155.7, 155.2, 138.8, 128.4, 128.0, 114.9, 110.5, 61.2. Anal. Calcd (%) for $C_{12}H_{12}N_6O_4S$: C: 42.85; H: 3.60; N: 19.03. Found: C: 42.86; H: 3.63; N: 19.04.

**General procedures for the synthesis of compounds (14a-g).** The compounds **14b** and **14e** were prepared according to the methodology previously described [21]. The compounds **14a**, **14c**, **14d**, **14f**, and **14g** were prepared according to the following general procedure: 1.61 mmol of 4-hydroxybenzaldehyde (**13**) was treated with previously selected hydrazides (**5a-e**; **6a-b**) (1.61 mmol) in 20 mL of ethanol, catalyzed by acetic acid (0.2 mL). The reactions were stirred at room temperature up to 24h and monitored by TLC until the reactive consumption. After, the solvent was partially removed under reduced pressure, followed by the addition of iced water (~ 15 mL) to precipitate the final *N*-acylhydrazones (**14a**, **14c**, **14d**, **14f**, and **14g**). The precipitate was collected by filtration and washed with cold water to provide the compounds with yields ranging from 34–81.4%. When necessary, further purification was performed using column chromatography (flash silica, eluent: 50% ethyl acetate; 50% hexane).

*(E)-2-(4-hydroxybenzylidene)hydrazine-1-carboxamide (14a)*. Off-white solid; yield 34%; mp: 224–226˚C. IR max (cm[-1]; KBr pellets): 3474 (O-H), 3258 (N-H), 1697 (C = O amide), 1604 (NH$_2$), 1580 (C = N imine), 1508 and 1451 (C = C), 1255 (C-O), 1169 (C-N). [1]H NMR (600 MHz, DMSO-$d_6$) δ: 10.02 (s, 1H), 9.73 (s, 1H), 7.73 (s, 1H), 7.52 (d, *J* = 8.7 Hz, 2H), 6.76 (d, *J* = 8.6 Hz, 2H), 6.37 (s, 2H). [13]C NMR (151 MHz, DMSO-$d_6$) δ: 158.4, 156.9, 139.6, 128.1, 125.8, 115.4. Anal. Calcd (%) for $C_8H_9N_3O_2$: C: 53.63; H: 5.06; N: 23.45. Found: C: 53.65; H: 5.08; N: 23.45.

*(E)-2-(4-hydroxybenzylidene)hydrazine-1-carbothioamide (14b)*. Off-white solid; yield 65%; mp: 227–229˚C. IR max (cm[-1]; KBr pellets): 3522 (O-H), 3198 (N-H), 1613 (C = O amide), 1587 (NH$_2$), 1549 (C = N imine), 1509 and 1451 (C = C), 1383 and 1281 (C = S), 1269 (C-O), 1165 (C-N). [1]H NMR (600 MHz, DMSO-$d_6$) δ: 11.25 (s, 1H), 9.87 (s, 1H), 8.06–7.83 (m, 2H), 7.95 (s, 1H), 7.61 (d, *J* = 8.7 Hz, 2H), 6.77 (d, *J* = 8.7 Hz, 2H). [13]C NMR (151 MHz, DMSO-$d_6$)

δ: 177.4, 159.2, 142.6, 129.0, 125.1, 115.5. Anal. Calcd (%) for $C_8H_9N_3OS$: C: 49.22; H: 4.65; N: 21.52. Found: C: 49.21; H: 4.65; N: 21.52.

*(E)-N'-(4-hydroxybenzylidene)benzohydrazide (14c)*. Off-white solid; yield 81%; mp: 244–245˚C. IR max (cm$^{-1}$; KBr pellets): 3194 (O-H), 3065 (N-H), 1608 (C = O amide), 1580 (C = N imine), 1564 and 1520 (C = C). $^1$H NMR (600 MHz, DMSO-$d_6$) δ: 11.65 (s, 1H), 9.94 (s, 1H), 8.35 (s, 1H), 7.90 (d, *J* = 7.2 Hz, 2H), 7.60–7.55 (m, 3H), 7.52 (d, *J* = 7.8 Hz, 2H), 6.84 (d, *J* = 8.6 Hz, 2H). $^{13}$C NMR (151 MHz, DMSO-$d_6$) δ: 162.8, 159.4, 148.1, 133.6, 131.5, 128.8, 128.4, 127.5, 125.3, 115.7. Anal. Calcd (%) for $C_{14}H_{12}N_2O_2$: C: 69.99; H: 5.03; N: 11.66. Found: C: 69.97; H: 5.02; N: 11.66.

*(E)-4-hydroxy-N'-(4-hydroxybenzylidene)benzohydrazide (14d)*. Off-white solid; yield 34%; mp: 224–226˚C. IR max (cm$^{-1}$; KBr pellets): 3522 (O-H), 3248 (N-H), 1632 (C = O amide), 1604 (C = N imine), 1508 and 1451 (C = C), 1290 (C-O). $^1$H NMR (600 MHz, DMSO-$d_6$) δ: 11.44 (s, 1H), 10.10 (s, 1H), 9.91 (s, 1H), 8.31 (s, 1H), 7.78 (d, *J* = 8.6 Hz, 2H), 7.54 (d, *J* = 8.2 Hz, 2H), 6.85 (d, *J* = 5.2 Hz, 2H), 6.82 (d, *J* = 5.1 Hz, 2H). $^{13}$C NMR (151 MHz, DMSO-$d_6$) δ: 162.6, 160.6, 159.3, 147.2, 129.6, 128.8, 125.5, 124.1, 115.7, 115.0. Anal. Calcd (%) for $C_{14}H_{12}N_2O_3$: C: 65.62; H: 4.72; N: 10.93. Found: C: 65.62; H: 4.74; N: 10.94.

*(E)-4-amino-N'-(4-hydroxybenzylidene)benzohydrazide (14e)*. Off-yellow solid; yield 65%; mp: 275–276˚C. IR max (cm$^{-1}$; KBr pellets): 3522 (O-H), 3248 (N-H), 1632 (C = O amide), 1604 (C = N imine), 1508 and 1451 (C = C). $^1$H NMR (600 MHz, DMSO-$d_6$) δ: 11.25 (s, 1H), 9.86 (s, 1H), 8.29 (s, 1H), 7.65 (d, *J* = 8.4 Hz, 2H), 7.51 (d, *J* = 8.3 Hz, 2H), 6.82 (d, *J* = 8.6 Hz, 2H), 6.58 (d, *J* = 8.7 Hz, 2H), 2.19 (s, 2H). $^{13}$C NMR (151 MHz, DMSO-$d_6$) δ: 162.8, 159.0, 152.1, 146.3, 129.2, 128.5, 125.7, 119.7, 115.6, 112.5. Anal. Calcd (%) for $C_{14}H_{13}N_3O_2$: C: 65.87; H: 5.13; N: 16.46. Found: C: 65.87; H: 5.15; N: 16.46.

*(E)-N'-(4-hydroxybenzylidene)-4-nitrobenzohydrazide (14f)*. Off-yellow solid; yield 75%; mp: 266–268˚C. IR max (cm$^{-1}$; KBr pellets): 3331 (O-H), 3191 (N-H), 1665 (C = O amide), 1594 (C = N imine), 1549 and 1440 (C = C), 1514 and 1340 ($NO_2$) 1267 (C-O). $^1$H NMR (600 MHz, DMSO-$d_6$) δ: 11.94 (s, 1H), 9.99 (s, 1H), 8.37 (s, 1H), 8.36 (d, *J* = 2.5 Hz, 2H), 8.14 (d, *J* = 8.5 Hz, 2H), 7.59 (d, *J* = 8.4 Hz, 2H), 6.85 (d, *J* = 8.2 Hz, 2H). $^{13}$C NMR (151 MHz, DMSO-$d_6$) δ: 161.1, 159.7, 149.2, 149.1, 139.3, 129.1, 125.0, 123.6, 115.7. Anal. Calcd (%) for $C_{14}H_{11}N_3O_4$: C: 58.95; H: 3.89; N: 14.73. Found: C: 58.95; H: 3.89; N: 14.74.

*(E)-4-(tert-butyl)-N'-(4-hydroxybenzylidene)benzohydrazide (14g)*. Off-white solid; yield 34%; mp: 224–226˚C. IR max (cm$^{-1}$; KBr pellets): 3522 (O-H), 3248 (N-H), 2961 (C-H, sp3), 1606 (C = O amide), 1577 (C = N imine), 1554 and 1511 (C = C), 1366 and 1305 ($CH_3$), 1278 (C-O), 1165 (C-N) $^1$H NMR (600 MHz, DMSO-$d_6$) δ: 11.59 (s, 1H), 9.95 (s, 1H), 8.33 (s, 1H), 7.83 (d, *J* = 8.3 Hz, 2H), 7.56 (d, *J* = 8.4 Hz, 2H), 7.53 (d, *J* = 8.3 Hz, 2H), 6.84 (d, *J* = 8.4 Hz, 2H), 1.31 (s, 9H). $^{13}$C NMR (151 MHz, DMSO-$d_6$) δ: 162.8, 159.4, 154.5, 147.8, 130.9, 128.8, 127.4, 125.4, 125.2, 115.7, 34.7, 30.9. Anal. Calcd (%) for $C_{18}H_{20}N_2O_2$: C: 72.95; H: 6.80; N: 9.45. Found: C: 72.97; H: 6.83; N: 9.45.

## Quantification of nitrite by the Griess reaction

The levels of nitrite resulting from the oxidation of NO in the aqueous medium were quantified through the Griess reaction after incubation with an excess of *L*-cysteine (1:50) according to previously published methods [23, 27, 28]. The experiments were performed in triplicate and repeated three times on different days. No production of nitrite was observed in the absence of *L*-cysteine. The results were expressed as a percentage of nitrite (% $NO_2^-$; mol/mol). Statistical analysis was carried out using ANOVA followed by Tukey's Multiple Comparison Test at a significance level of $p < 0.05$.

## Cloning and expression of CPB in the yeast *Pichia pastoris*

*Escherichia coli* TOP 10 (Invitrogen) was used as a host for gene cloning experiments, whilst the yeast *Pichia pastoris* X-33 (Invitrogen), for the expression of CPB2.8 (CPB). The ORF that codes for the *Leishmania mexicana* CPB, lacking the C terminal extension (CPB2.8ΔCTE) [29], was PCR amplified using the *Taq* polymerase HiFi (Cellco Biotecnologia, São Carlos, SP) and the pair of primers (5´–AAA**GAATTC**GCCTGCGCACCTGCGCG–3´) forward and (5´–AAA**GTCGAC**CCGCACATGCGCGGACACGG–3´) reverse, which contains the restriction sites *Eco*RI and *Sal*I, respectively. The PCR product was purified from 1% agarose gel and cloned into pGEM-T easy vector (Promega), following the manufacturer's instructions and used to transform TOP10 *Escherichia coli* (*Invitrogen*). After identification and production of the recombinant plasmid, it was double digested with *Eco*RI and *Sal*I, the band of ~1 kbp was isolated from 1% agarose gel and ligated to the *Eco*RI/*Sal*I digested pPICZαA vector (Invitrogen) to produce a recombinant protein bearing a C-terminal histidine tag. The obtained recombinant vector was confirmed by sequencing using the primers forward *α factor* (TAC TATTGCCAGCATTGCTGC) and reverse 3'AOX1 (GCAAATGGCATTCTGACATCC) and hereafter, linearized with *Pme*I for the transformation of *P. pastoris* X-33 (Invitrogen) by electroporation in a Gene-Pulser (Bio-Rad, US) at Gene Pulse X-Cell (*Bio Rad*), 1,5 KV, 25 μF e 200 Ω, as described [30]. The cells were plated onto YPDS-agar medium (1% yeast extract, 2% peptone, 2% dextrose, 1 M sorbitol, 2% agar) containing 100, 250 and 500 μg.mL$^{-1}$ of Zeocin™, and incubated at 30˚C for 3–5 days for selection of zeocin resistant colonies, which were analyzed by PCR to confirm the presence of the *Eco*RI/*Sal*I fragment into the yeast genomic DNA. Twenty-four transformants were screened for recombinant protease production after methanol induction in 24-well plates (*Whatman*) as previously described [31] with modifications in the cultivation and induction [30], and the protease production monitored in 15% (w/v) SDS-PAGE [32]. The clone that better produced rCPB2.8ΔCTE, observed by SDS-PAGE, was induced again in 100 mL of medium, for 96 h for purification. The recombinant enzyme was purified by immobilized metal affinity chromatography (IMAC) using Ni-NTA agarose resin Superflow (Qiagen®) and eluted using a buffer containing different concentrations of imidazole (25 mM to 250 mM) 10 mM Tris, 50 mM Na$_2$HPO$_4$ monohydrate, and 100 mM CaCl$_2$. After elution, dialysis of the fractions containing the recombinant protein was carried out in SnakeSkin Dialysis Tubing, 3.5K MWCO (Thermo Scientific) against 100 mM sodium acetate buffer pH 5.0 (2L), 2X, and the protein quantified using the *Pierce BCA Protein Assay Kit* (Thermo Fisher Scientific).

## Enzymatic assays

The concentration of the active recombinant CPB expressed in *Pichia pastoris* supernatant was determined by active site titration with *N*-(trans-Epoxysuccinyl)-L-leucine 4-guanidinobutylamide E-64 (CalBiochem) [33] that was found 1 μM. The active enzyme represented about 80% of the total recombinant enzyme.

Compounds were dissolved in DMSO to a final concentration of 20 μM. The assays were performed in dark 96 well-plates, containing 200 μL of 100 mM sodium acetate buffer (pH 5.5), 200 mM NaCl, 0.01% Triton-X100, 2 nM enzyme, various concentrations of inhibitor, and 5 μM of the substrate Z-FR-AMC (Calbiochem). The final concentration of DMSO was 1% (v/v). The enzyme was pre-incubated at 30˚C for 2 minutes before the inhibitor was added, waited 2 minutes and after this incubation, the reaction was started by the addition of the substrate, Z-FR-AMC. The fluorescence was monitored at 380 nm excitation and 460 nm emission filters for 2 minutes at 30˚C in an Infinite 200 PRO microplate reader (Tecan, Männedorf, Switzerland). The residual activity was measured by the ratio between enzyme + substrate and enzyme

+ inhibitor + substrate. All compounds were evaluated at 20 μM, and the compounds able to inhibit the enzyme at 20 μM (inhibition < 20 μM) were selected and tested again at 10 μM, 5 μM, 2.5 μM, 1.25 μM, 0.62 μM, and 0.3 μM to determine $IC_{50}$ values [16]. Results were expressed in mean ± standard deviation of two independent replicates using Bioestat Software.

## Biological assays

**Parasites and cells.** Promastigotes of *L. infantum* strains MHOM/BR/1972/LD and MHOM/MA/67/ITMAP-263 were maintained at 27˚C in Schneider's medium (Sigma) supplemented with 100 mL of heat-inactivated fetal bovine serum (FBS), streptomycin, and penicillin (100 mg $mL^{-1}$; 100 μM $mL^{-1}$) (LGC Biotecnologia) and 10% of sterile masculine human urine, in 25 $cm^2$ culture bottles (TPP). Peritoneal macrophages were obtained as previously described [34]. Briefly, cells were collected from the peritoneal cavity of six to eight weeks old male Swiss mice, previously stimulated with thioglycolate 3%. For this, PBS (pH 7.4) was injected into the peritoneal cavity, followed by a slight massage, and the content was collected using a syringe (5 mL). These cells were cultivated in RPMI 1640 medium supplemented with 10% heat-inactivated FBS, 25 mM HEPES, 2 mM L-glutamine, 1% penicillin/streptomycin and incubated at 37˚C in a 5% $CO_2$-air mixture in 96 well-plates or 24 well-plates.

**Evaluation of cytotoxicity on peritoneal macrophages.** In summary, cells (5 x $10^5$ cells $mL^{-1}$) were incubated in RPMI 1640 medium, containing decreasing concentrations of compounds (500 to 7.8 μM), at a final volume of 100 μL per well, for 24 h at 37˚C in a 5% $CO_2$-air. Their viability was measured by the MTT colorimetric assay according to Velasquez et al. (2016) [34] with some modifications. Briefly, after the incubation above described, the supernatant was removed and it was added 100 μL of ethanol and 100 μL of PBS: isopropanol (v/v). The absorbance was read on a spectrophotometer at 570 nm. The cytotoxic concentration of compounds that resulted in 50% of cell growth inhibition ($CC_{50}$) was determined using non-linear regression on Bioestat® software. The assays were performed in experimental triplicate and two independent replicates. Results expressed in mean ± standard deviation.

**Anti-amastigote activity.** The antileishmanial activity against *L. infantum* intracellular amastigotes, according to the following methodology [34]. Murine peritoneal macrophages in RPMI 1640 were seeded at a density of 5 x $10^5$ cells $mL^{-1}$ (500 μL final volume) containing coverslips of 13 mm diameter arranged in a 24-well plate. After 6 h of incubation at 37˚C and 5% $CO_2$-air mixture for macrophage adhesion, promastigote forms of *L. infantum* at the stationary growth phase (6–7 days) were added to the wells in a ratio of 10:1 (promastigotes: macrophages) and incubated for another 18 h to allow parasites internalization. After incubation, the medium was replaced, non-internalized parasites were removed by PBS washing and added a new medium containing 10 μM to 0.25 μM of compounds and plates were incubated for 24 h at 37˚C and 5% $CO_2$-air mixture. Were used infected macrophages as negative control and amphotericin B as the positive control.

After incubation, cells were fixed with methanol and Giemsa stained. The effective concentration to 50% amastigotes death (EC) was obtained by counting 100 macrophages in duplicate, in two independent experiments. The values were obtained by non-linear regression on Bioestat® software, and results were expressed in mean ± standard deviation. Additionally, was determined the Selective Index (SI = $CC_{50macrophages}/EC_{50leishmania}$) indicating how many times the sample is more selective to parasites.

***In vivo* analysis.** To evaluate the *in vivo* antileishmanial efficacy of compound **4f**, male BALB/c mice, 20 ± 4 g, 6 weeks-old (CEMIB-UNICAMP) were intraperitoneally inoculated with 100 μL (1x$10^8$ parasites) of infective promastigotes of *L. infantum* (MHOM/MA/67/ITMAP-263) at the stationary phase. After nine days post-infection, the animals were

treated by oral gavage (100 μL per dose) twice daily for 5 days with a 12h interval between doses according to the dosages described below, and euthanized [35, 36]. The adopted doses, obtained through extrapolation based on $EC_{50}$ and mice blood volume/ body weight [37]; supplementary material were: (0.28 mg/kg/day, 0.85 mg/kg/ day); 2.56 mg/kg/ day; 7.7 mg/Kg/ day, the same used for the reference drug, miltefosine (Cayman chemical) previously reported (7.7 mg/kg/day) [38]. Work solutions were daily prepared in PBS. Negative controls correspond to infected and non-treated mice. As a positive control, a group of mice received 7.7 mg/kg/day of miltefosine, and a group of uninfected and untreated mice (healthy animals) was used for the analysis. The compound and miltefosine were orally administered twice a day for five days. After treatment, the animals were euthanized and the treatment efficacy was assessed by measuring the parasite burden of the infected animals using the LDU index [38, 39].

*LDU index.* After the euthanasia, mice´s liver and spleen were collected for parasite quantification. Tissues impression onto microscopy slides were made, followed by Giemsa-staining and examining under optical microscopy. The corresponding number of amastigotes per 1,000 nucleated cells/organ weight was expressed as LDU index according to Stauber (1958), Sousa-Batista (2018), and Kwofie (2019) [38–40].

*Toxicity for mice assays.* The blood was collected in heparinized tubes and immediately centrifuged at 700×*g* for 10 min at 25˚C to obtain plasma. Plasma levels of aspartate (AST), and alanine (ALT) transferases, alkaline phosphatase (ALP), creatinine, urea, and total bilirubin were determined at the end of treatment using commercial kits (Labtest Diagnóstica S.A., Brazil). The assays were performed according to Velasquez et al. (2017) [37] and the data are expressed as average ± SEM. The statistical differences between groups were evaluated using a one-way analysis of variance, followed by the Student-Newman-Keuls multiple comparison test using GraphPad Prism software. Differences were considered significant when P values were 0.05.

**Ethics statement.** All experiments involving animals were approved by the Ethics Committee for Animal Experimentation of School of Pharmaceutical Sciences, Sao Paulo State University (CEUA-FCF-UNESP), under the number (CEUA/FCF/CAr:17/2020), in agreement with the guidelines of the Sociedade Brasileira de Ciência de Animais de Laboratorio (SBCAL) and the National Council for the Control of Animal Experimentation (CONCEA). Euthanasia was performed using xylazine/ketamine anesthesia, followed by cervical dislocation. All necessary efforts were taken in order to minimize animals discomfort and pain.

## Computational methods

*In silico* **prediction of ADME properties.** Determination of the potential of oral absorption, drug-likeness, and water solubility of the furoxan derivatives was performed using the Swiss ADME software [41].

**Molecular docking.** All *in silico* studies were performed on the Maestro molecular modeling environment. The 3D structures were generated applying the *LigPrep* procedure. It was used *Epik* [42] to generate the protonation state in pH 5.5 ± 1.0 (the same used on CPB enzymatic assays) and OPLS03 as a field force. The coordinate for the target (PDB ID: 6P4E), resolution: 1.35 Å [43] was retrieved from Protein Data Bank. The .*pdb* structure was prepared using *ProteinPreparationWizard*. The hydrogens were removed and reincluded. The missing residues were corrected. All residues were optimized based on pH 5.5 ± 1.0. The minimization steps run until the converge threshold RMSD equals 0.15 Å. The grid generation was prepared with volume appropriated to cover all investigational sites. The catalytic amino acids CYS26 and HIS164 were considered as deprotonated and protonated, respectively. Before grid

generation, it was performed as default Protein Preparation Wizard. It generated a grid box of 15 Å centered on catalytic cysteine. All docking calculations were performed using *Glide* on *Extra-precision* mode (XP) [44]. For all docking simulations were generated 20 poses and post-docking minimization. Other parameters were kept as default.

## Results

### Chemistry

The phenyl-furoxan (**3**) was obtained in two steps, with an overall yield of 84% (Scheme 1). First, the styrene (**1**) was treated with sodium nitrite and acetic acid at room temperature in the medium containing dichloromethane as solvent to obtain the furoxan derivative (**2**) at 87%, as previously described [23]. Then, compound (**2**) was treated with 4-hydroxybenzaldehyde and 1,8-diazabicyclo [5.4.0] undec7-ene (DBU) in a dichloromethane medium to provide compound (**3**) at a yield of 90% [21]. The amide-furoxan derivative (**9**) and benzofuroxan derivative (**12**) were obtained according to the previously described procedures [24, 25]. Finally, the last step involved the coupling reaction of the aldehyde group in phenyl-furoxan (**2**), amide-furoxan (**9**), or benzofuroxan (**12**) within the appropriate hydrazides (**5a-e**; **6a-b**) in medium containing ethanol and acetic acid at yields ranging from 82 to 96%, providing the *N*-oxide compounds (**4a-x**) (Schemes 1 and 2).

In order to evaluate the contribution of *N*-oxide subunit, the *N*-acylhydrazone analogs (**14a-g**) were prepared through condensation reaction involving 4-hydroxybenzaldehyde and distinct hydrazides (**5a-e**; **6a-b**) in a medium containing ethanol and acetic acid with yields ranging from 82 to 96% (Scheme 2). The structure and the purity (>98.5%) of all final compounds were determined, as well as their $^1$H NMR spectra (S1 Appendix), which revealed a single signal corresponding to ylidenic hydrogen of the *E*-diastereomers.

**a)** NaNO₂, acetic acid, HCl, dichloromethane, r.t., 12h; **b)** 4-hydroxybenzaldehyde, DBU, dichloromethane, r.t., 12h; **c)** respective hydrazide, ethanol, acetic acid (cat), r.t., 12h; **d)** thionyl bromide, DMF, r.t., 30 min; **e)** 4-hydroxybenzaldehyde, DBU, dichloromethane, r.t., 1h.

**Scheme 1. Synthesis of the furoxan derivatives (4a-o).**

**Scheme 2. Synthesis of benzofuroxan (4p-x) and N-acyl-hydrazones (14a-g) derivatives.**

## Nitrite measurement

The ability of all synthesized compounds (**4a-x**) and (**14a-g**) to release NO was evaluated by indirect determination of $NO_2^-$ production. The compounds at $10^{-4}$ M were incubated for 1 h at 37˚C in the presence of *L*-cysteine (1:50) and the produced $NO_2^-$ was determined through Griess reaction (Table 1) [23].

The furoxan derivative (**Lapdesf14e**), which contains a phenylsulfonyl moiety attached to C-3 was more prone to release NO (26%) [21], whilst those containing phenyl (**4a-g**) (9%), or amide (**4h-o**) (12%) groups at the same position exhibited a similar pattern of nitrite formation when compared to DNS, used as reference NO donor (10.5%). On the other hand, both benzo-furoxan (**4p-x**) and *N*-acylhydrazones (**14a-g**) were not able to release NO.

## Recombinant expression of CPB

The recombinant cysteine protease CPB was, for the first time, successfully expressed in *P. pastoris*, circumventing the problem of solubility and yield when expressed in *Escherichia coli* [29]. The CPB coding sequence was PCR amplified from *Leishmania mexicana* genomic DNA, excluding the C-terminal extension region, which is not necessary for its activity against Trypanosomatids, as previously described [45, 46] and later confirmed by Sanderson and colleagues [29]. After sequence confirmation (S1 Fig), the plasmid containing the CPB coding region, named pPIC-CPB2.8ΔCTE, was used to transform the *P. pastoris* X-33 strain. Among the positive clones, the pPIC-CPB2.8#25 was selected for production of CPB in medium containing methanol. The culture supernatants from 0h to 144 h were collected and visualized in 15% SDS-PAGE, stained with Coomassie Brilliant Blue R-250 to verify the expression levels (Fig 2A).

Protein production started at 24 h and increased until 144 h after methanol induction. At 24 h and 48 h, it is observed bands above 30 kDa that disappear after 72 hours, probably due to medium acidification, leading to the loss of the pro-peptide and conversion of the enzyme to its active form around 26 kDa [29]. Therefore, this clone was chosen for scaling-up production (100 mL) at 96 h, followed by protein purification.

Table 1. NO-released data for compounds (4a-x) and (14a-e).

| Compounds ($10^{-4}$ M) | % NO$_2^-$ (mol/mol) |
|---|---|
| | 5 mM of *L*-Cys |
| DNS | $10.5 \pm 0.9^{*a}$ |
| Lapdesf14e | $26.0 \pm 1.6^{*\dagger}$ |
| 4a | $9.0 \pm 0.7^*$ |
| 4b | $8.8 \pm 0.6^*$ |
| 4c | $9.3 \pm 0.8^*$ |
| 4d | $8.6 \pm 0.8^*$ |
| 4e | $8.8 \pm 0.7^*$ |
| 4f | $9.1 \pm 0.7^*$ |
| 4g | $9.0 \pm 0.6^*$ |
| 4h | $12.1 \pm 1.2^*$ |
| 4i | $12.3 \pm 0.9^*$ |
| 4j | $12.7 \pm 0.8^*$ |
| 4l | $11.8 \pm 1.1^*$ |
| 4m | $12.5 \pm 1.3^*$ |
| 4n | $12.0 \pm 1.4^*$ |
| 4o | $11.8 \pm 1.2^*$ |
| 4p-4x | 0 |
| 14a-14g | 0 |
| AmpB | 0 |

[a]DNS: isosorbide dinitrate (DNS possesses two ONO$_2$ groups that may release NO). The data are expressed as the means ± standard errors of the means. Significant differences between the experimental and control groups were evaluated by analysis of variance followed by Tukey's Multiple Comparison Test.

* $p < 0.05$ *vs.* ampB.

† $p < 0.05$ *vs.* DNS.

The clone pPICZαA#25 was cultivated for 96 h, and the purification of CPB was performed according to the EasySelect *Pichia* Expression kit (*Invitrogen*) and further analyzed by 15% SDS-PAGE. The protein was eluted at imidazole 75 mM, presenting a major band around 26 kDa, although low levels of protein were also detected in all concentrations of imidazole (Fig 2B). After quantification, a total of 40 mg L$^{-1}$ of protein was obtained.

The expected size for the fully active form of the protein, approximately 26 kDa (without the pro-peptide) was observed after dialysis. The additional bands might be the result of auto-proteolysis leading to the auto-cleavage of the *N*-terminus, i.e., the proteolysis products that are fused to the His-tag were purified in the column. Thus, to circumvent the auto-proteolysis process and to determine the percentage of active enzyme present in the purified sample, an active site titration was performed using the cysteine protease inhibitor, E-64 (Fig 2C), according to Barrett et al. (1981) [33], showing that the percentage of the active enzyme was 80%, corresponding to 32 mg L$^{-1}$ of purified recombinant and active enzyme.

## Antileishmanial activity and inhibition of CPB by *N*-oxide derivatives

Furoxan (4a-4o) and benzofuroxan (4p-4x) antileishmanial activities against *L. infantum* intracellular amastigotes as well as their inhibitory effect against CPB were investigated.

The furoxan derivatives (4a-4h, 4l and 4m) and the benzofuroxan (4r), exhibited anti-amastigote potency (EC$_{50}$) lower than < 10 μM. These molecules were also able to target the CPB at different levels ranging from 0.8 to 8.8 μM (Table 2). Moreover, except for 4c and 4r,

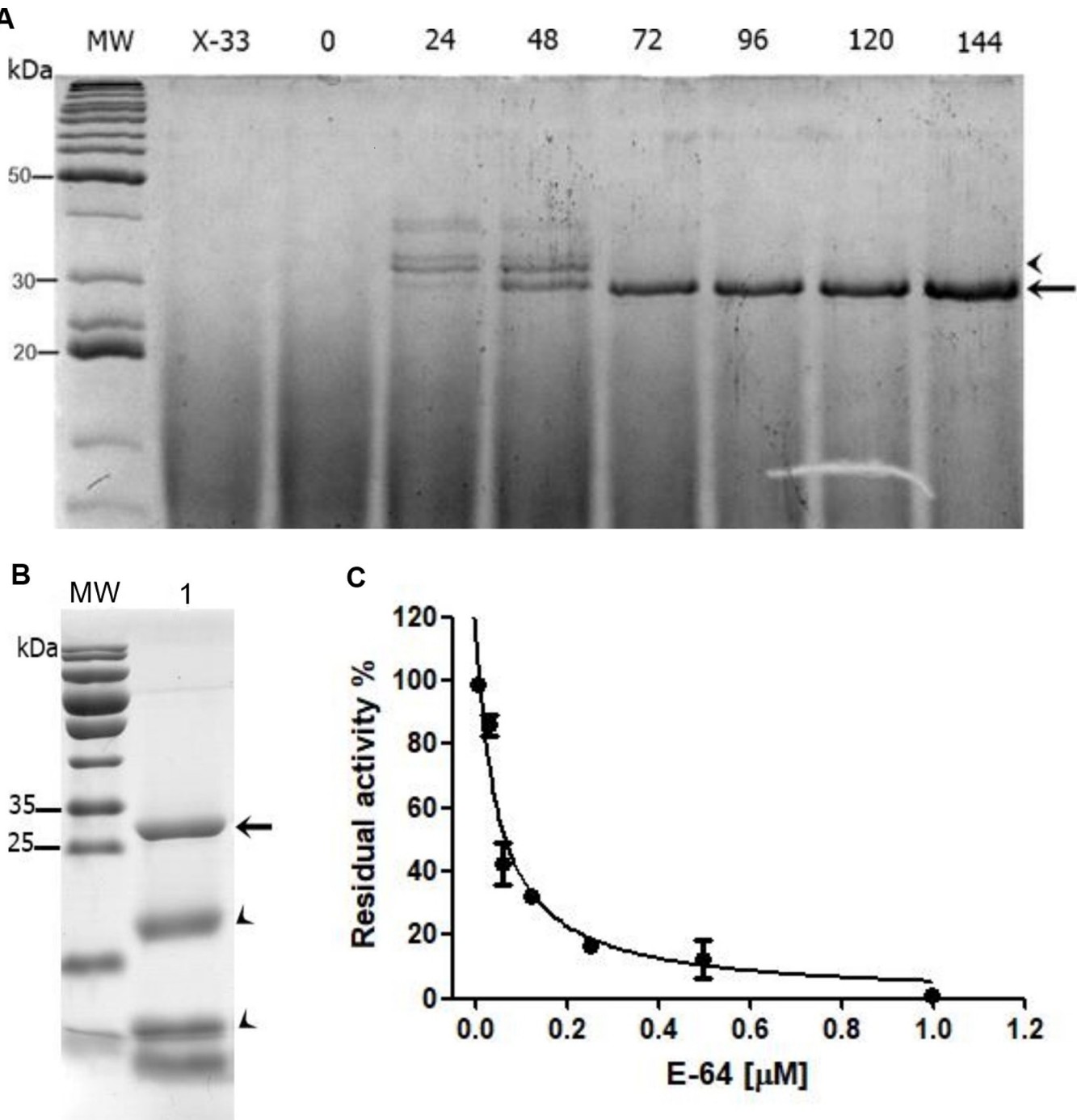

**Fig 2. Analysis of CPB expression in *P. pastoris*.** (A) SDS-PAGE showing the supernatants of non- transformed *Pichia pastoris* X-33, and aliquots from 0 to 144 hours of the transformed clone pPIC-CPB2.8#25 induced with methanol. The arrow indicates the expected size for active (processed) CPB. The arrowhead indicates the unprocessed protein after 24 and 48 h induction. MW: molecular weight Bench Mark (Invitrogen). (B) Analysis of *L. mexicana* CPB after purification and dialysis. Coomassie-stained SDS-PAGE 15% showing in MW: molecular weight Page Ruler (Thermo Scientific); 1, the purified and dialyzed recombinant CPB. The arrow indicates the recombinant protein (about 26 kDa) and the arrowheads, the products of proteolysis. (C) Active site titration of the recombinant CPB using the irreversible protease inhibitor E-64. Residual activity % (Relative Fluorescence Units) was measured by the ratio between the enzyme activity in absence or presence of increasing concentrations of E-64.

the cytotoxicity of the aforementioned compounds was at least 50 times more selective to the parasite rather than to the macrophages, considering the SI values (Table 2). Interestingly, although the furoxan (**4j** - **4n**) and benzofuroxan (**4q**, **4u**, **4t**) inhibited the recombinant enzyme, they did not affect the parasite viability.

To provide comprehensive proof of concept regarding the effect of the *N*-acylhydrazone on CPB, a series of hydrazone derivatives (**14a-14e**) was also evaluated. Although none of them exhibited antileishmanial activity (EC$_{50}$ >10 μM), they were able to inhibit CPB at different IC$_{50}$ levels, ranging from 1.2 to 11.7 μM.

## Molecular docking analyses

Molecular docking was carried out to comprehend the pose of synthesized furoxan (**4a-o**), benzofuroxan (**4p-x**) and hydrazone derivatives (**14a-g**) into the CPB active site. The docking results of the *N*-acylhydrazones series (**14a-g**) suggested an interaction between the N-H of the compounds and the CYS26 residue of the catalytic site (Fig 3A). Except for hydrazone **14e,** which contains a bulky group, similar poses were observed for all other *N*-acylhydrazones. The contribution of the *N*-oxide group of the furoxans and benzofuroxans (**4a-x**) was orientated on the S1' pocket, performing an interaction between a cation-π and the TRP186 residue (Fig 3B and 3C). Additionally, the imine carbon (C = N) of the *N*-acylhydrazone moiety (**4a-x**) has been positioned near the CYS26 at an average distance of 3.56 Å and the N-H of their scaffolds interacts with ASN163 residue through an H-bond (Fig 3B and 3C).

Superposition of **14a**, **4a,** and **4e** revealed an occupancy of the S2 pocket by the phenyl ring of the phenylfuroxan derivatives (**4a-g**) (Fig 3). Besides, a hydrophobic interaction between the furoxan **4e** and the S2 pocket, not observed for the hydrazone **14a**, might explain the two-fold higher inhibitory enzyme activity exhibited by the furoxan. The improvement on the occupancy by phenylfuroxan derivatives (**4a-g**) in the CPB can also explain the better inhibitory effect of this series compared with its respective hydrazones (**14a-g**) (Fig 3).

## *In silico* ADME studies

In an attempt to select the best hit among the most potent and selective compounds (**4a**-**h**, **l**, and **m**), we performed *in silico* analyses using the SwissADME tool [41] to investigate physicochemical properties, pharmacokinetic parameters, and drug-likeness (S1 Table). Amide compounds exhibiting moderate to high solubility, **4a-c** and **4f** displayed better drug likeness by complying with Lipinski rule of five and the filters Ghose, Veber, Egan, and Muegge in contrast to **4d**, **4e**, **4g**, **4h**, **4l,** and **4m**, which violates some of the properties/descriptors. The analysis of potential binding to main cytochrome CYP450, a superfamily of liver isoenzymes responsible for the metabolization of xenobiotics, suggests a non-promiscuous behavior of compounds **4f**, **4h,** and **4l** since they show potential to inhibit only one isoenzyme (CYP1A2 or CYP2C19).

Hence, considering that **4f** combines both relevant bioactivity properties and the most favorable ADME parameters, highlighting its potential gastrointestinal absorption potential, this compound was selected for further *in vivo* studies to evaluate its potential efficacy and toxicity in *L. infantum* infected mice after oral delivery.

## *In vivo* studies

**Parasite load.** The compound **4f** was investigated concerning its capacity to reduce parasite load in both liver and spleen of *L. infantum*-infected Balb/C mice. Therefore, nine days post-infection, **4f** at four different doses (0.28, 0.85, 2.56, and 7.7 mg/kg/day) or the reference drug miltefosine (7.7 mg/kg/day) were orally administered twice a day for five days. Parasite

**Table 2. Inhibition of CPB, biological activities, and safety by furoxan and benzofuroxan derivatives (µM).**

| | Compound | $IC_{50}$ | $EC_{50}$ | $CC_{50}$ | SI |
|---|---|---|---|---|---|
| **Furoxan** | **Lapdesf14e** | 3.1 ± 0.4 | 3.1 ± 0.1 | 208.3 ± 0.2 | 66.4 |
| | **4a** | 1.5 ± 0.1 | 1.4 ± 0.1 | > 500 | 357 |
| | **4b** | 3.2 ± 0.3 | 3.1 ± 0.2 | 166.0 ± 0.5 | 54 |
| | **4c** | 2.0 ± 0.1 | 7.2 ± 0.2 | 122.0 ± 20.0 | 17 |
| | **4d** | 1.8 ± 0.1 | 3.9 ± 0.4 | > 500 | 128 |
| | **4e** | 0.8 ± 0.1 | 0.6 ± 0.1 | > 500 | 833 |
| | **4f** | 4.5 ± 0.1 | 3.6 ± 0.1 | > 500 | 139 |
| | **4g** | 6.5 ± 1.0 | 2.2 ± 0.3 | > 500 | 227 |
| | **4h** | 8.8 ± 0.2 | 3.1 ± 0.2 | > 500 | 161 |
| | **4i** | 5.7 ± 0.4 | > 10 | > 500 | - |
| | **4j** | 11.1 ± 0.1 | > 10 | > 500 | - |
| | **4l** | 10.8 ± 2.3 | 7.7 ± 2.1 | > 500 | 65 |
| | **4m** | 6.0 ± 0.7 | 6.4 ± 1.1 | > 500 | 78 |
| | **4n** | 13.7 ± 1.8 | > 10 | > 500 | - |
| | **4o** | > 20 | > 10 | > 500 | - |
| **Benzofuroxan** | **8a**[*] | > 80 | 18.2 ± 0.1 | 63.2 | 3.5 |
| | **4p** | > 20 | > 10 | > 500 | - |
| | **4q** | 14.0 ± 2.0 | > 10 | > 500 | - |
| | **4r** | 3.4 ± 0.1 | 9.5 ± 0.6 | 103.2 ± 2.7 | 11 |
| | **4s** | > 20 | > 10 | 152.0 ± 18.0 | - |
| | **4t** | 4.2 ± 0.2 | > 10 | 17.2 ± 0.7 | - |
| | **4u** | 11.7 ± 0.5 | > 10 | 265.0 ± 5.0 | - |
| | **4v** | > 20 | > 10 | > 500 | - |
| | **4x** | > 20 | > 10 | 246.2 ± 0.2 | - |
| **N-acylhydrazone** | **14a** | 2.0 ± 0.1 | > 10 | > 500 | - |
| | **14b** | 11.7 ± 1.0 | > 10 | > 500 | - |
| | **14c** | 2.0 ± 0.3 | > 10 | > 500 | - |
| | **14d** | 2.0 ± 0.1 | > 10 | > 500 | - |
| | **14e** | 1.2 ± 0.1 | > 10 | > 500 | - |
| | **14f** | 4.0 ± 0.1 | > 10 | > 500 | - |
| | **14g** | 7.6 ± 2.1 | > 10 | > 500 | - |
| **Control** | **Amphotericin B** | - | 1.0 ± 0.1 | 12.1 ± 1.8 | 12 |

$IC_{50}$—Inhibition of CPB. $EC_{50}$—**E**ffective **C**oncentration that kills 50% of *L. infantum* amastigote forms. $CC_{50}$—**C**ytotoxic **C**oncentration to kill 50% of murine peritoneal macrophages, all compounds were evaluated from 7.5 to 500 µM. SI—**S**elective **I**ndex ($CC_{50}/EC_{50}$) [20, 21].

load evaluation showed that **4f** (0.28 mg/kg/day) was able to reduce the number of amastigotes in both organs by 67–68% (Fig 4A and 4B), whilst the highest **4f** dose (7.7 mg/kg/day) induced a reduction similar to that obtained for the reference drug, i.e., ~90% in both spleen and liver.

*In vivo* **quantification of biomarkers.** The plasma levels of AST, ALP, ALT, total bilirubin, and creatinine were monitored to investigate liver and renal functions in healthy and *L. infantum*-infected mice treated (different doses of **4f** or miltefosine) and control. Total bilirubin levels in *L. infantum*-infected mice increased in contrast to the healthy ones (S2 Fig), whereas no significant alterations were observed for both **4f** and miltefosine treatment in comparison to the infected animals. Besides, no alterations were observed for any group to creatinine (S2 Fig). The levels of AST, ALP, ALT, and creatinine were similar in both treated and healthy groups (S2 Fig).

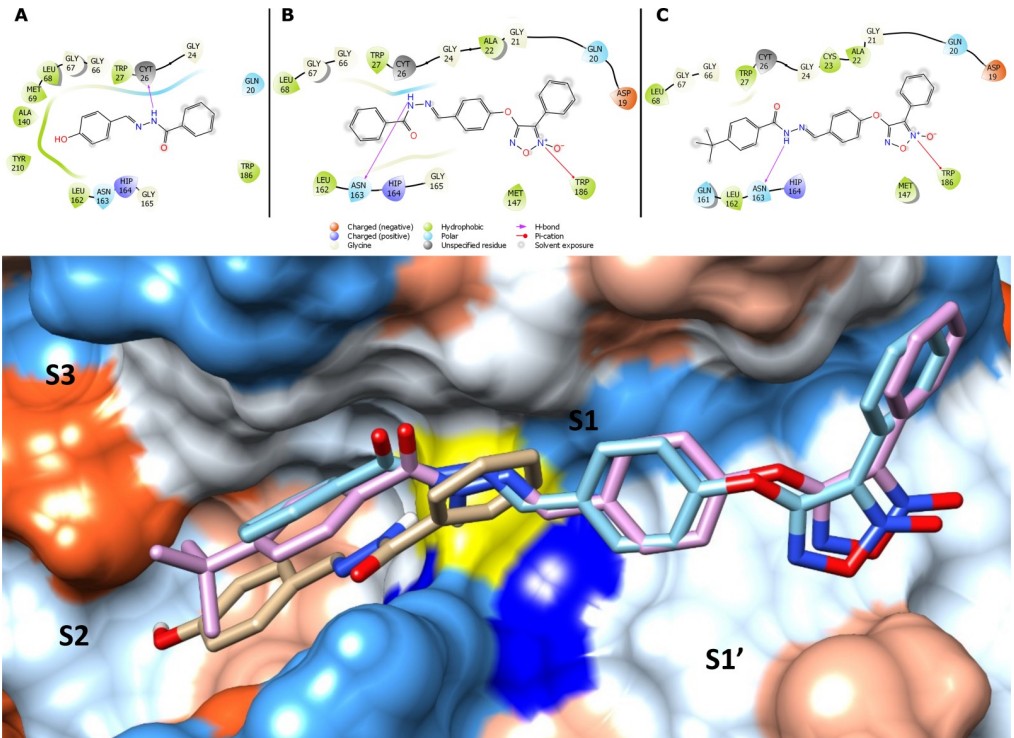

**Fig 3. Ligand-CPB complex predicted by molecular docking.** Possible poses and interactions performed by compounds **14a (A)**, **4a (B)** and **4e (C)** against CPB. Superposition of **4a** (beige), **14a** (cyan) and **14e** (pink). The yellow surface represents CYS26 and dark blue is HIS164.

## Discussion

The current antileishmanial therapy exhibits several drawbacks, including long-term treatment, adverse side effects, low efficacy, and difficult route of administration, making urgent the need for more effective and safer treatments. Herein, we described the synthesis and antileishmanial properties of new *N*-oxide derivatives, developed based on the prototype **Lapdesf14e** focusing on a dual effect: CPB inhibition and NO-release [20, 47]. The herein explored CPB is mainly expressed in amastigotes and has been assigned as a validated target for *Leishmania* drug discovery due to its important role in parasite escape from the host immune system, thus favoring parasite establishment and proliferation [8, 48, 49].

The CPB consists of active sites comprising pockets or subsites known as S1, S2, S3 (along with the catalytic cysteine to the C-terminal region), and S1' (along with the catalytic cysteine to the N-terminal region), interacting with the substrates P1, P2, P3 and P1' [50, 51]. Indeed, several CPB inhibitors have been described, including both natural products [52–55] and synthetic compounds [12, 16, 56, 57]; however, none of them are *N*-oxide derivatives. The performed molecular docking analyses suggested that the furoxan *N*-oxide subunit interacts with the S1' pocket through a cation-π interaction with the TRP186 residue. Noteworthy, for the phenyl furoxan derivatives (**4a-g**), the S2 pocket is occupied by a phenyl ring attached to C-3, which in association to the hydrophobic nature, might be positively contributing to the observed levels of enzyme inhibition ranging from 0.8 to 6.5 μM in contrast to the lower inhibition activity (6 to >20μM) related to the amide furoxans (**4h-o**). Indeed, Fey et al. (2018) also reported the binding nature of the aziridine-containing phenylpropil ester groups to the S2 pocket in association with studies of mechanisms of enzyme inhibition [13]. Furthermore,

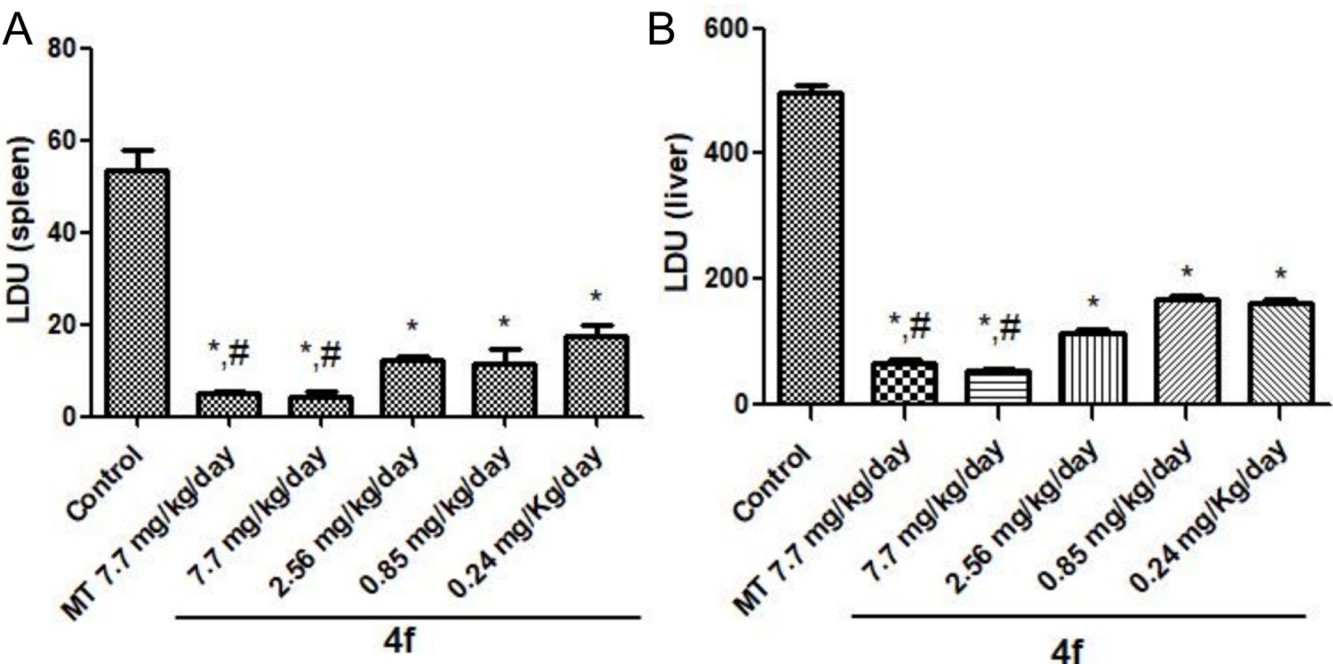

**Fig 4. Parasite load evaluation.** The LDU index (number of *Leishmania* amastigotes in 1,000 nucleated cells per organ weight) was determined in the spleen (**A**) and liver (**B**) of *L. infantum* infected BALB/c mice using. The data are expressed as average ± SD. Control: Infected and non-treated animals. MT: miltefosine. *: Statistically significant difference in all groups compared to the Infected and non-treated animals (p < 0.0001). #: Not statistically significant (p > 0.05).

the benzofuroxans (**4q, 4r, 4t, 4u**) also interact with the S1 and S1´pocket, contributing to the enzyme inhibition at different levels (3.4 to 14 μM).

Hereafter, the enzymatic inhibitory properties of the *N*-oxide derivatives were assessed using the recombinant enzyme CPB, which was alternatively expressed in *P. pastoris*, overcoming solubility issues commonly associated with the purified enzyme obtained in bacteria [29].

Although other studies suggested that protease inhibition might be caused by nitrosylation of cysteine residues of CPB [58–60], the NO-releasers phenyl and amide furoxans (**4a-o**) derivatives herein evaluated displayed corresponding levels of CPB inhibition ($IC_{50}$ ranging from 0.8–13 μM) to those *N*-acylhydrazone derivatives (**14b-g**) that do not present NO-release capacity ($IC_{50}$ ranging from 1.2 μM– 7.6 μM). Thus, besides the pharmacodynamics improvements based on **Lapdesf14e** allied to the inhibition of the targeted protease, we cannot exclude a contribution of the NO to the antileishmanial activity due to a pleiotropic effect caused by nitrosylation of other parasite proteins/enzymes rather than the cysteine CPB, which might be inducing metabolic/ signaling pathway disruption, ultimately contributing to parasite death [21, 47, 61]. Indeed, all the furoxans derivatives, except **4i, 4j,** and **4n**, presented anti-*L. infantum* amastigote activity lower than 10 μM.

Although there are several natural and synthetic CPB inhibitors reported [12, 16, 43–48], none of them have been further preclinical evaluated in visceral leishmaniasis models, except for compound K11777, which entered the clinical trials for Chagas disease and was later abandoned due to safety issues [11]. On the other hand, there are few oral antileishmanial compounds active against visceral leishmaniasis at phase I of clinical studies, and at this moment, none of them have their mechanisms of action described [62], which turns difficult further improvements based on medicinal chemical approaches.

Thus far, infected mice treated with **4f** at 7.7mg/kg/day was able to decrease the parasite load by 90%, consistent with miltefosine in the experimental conditions of this work and accordance with the hit-to-lead criteria established to the drug discovery field of neglected diseases [63]. Furthermore, even at a dose three times lower (2.56 mg/kg/day), the parasite reduction was 77% (0.28 mg/kg/day), successfully indicating a robust reduction of the parasite burden at the target organs in mice, without causing any evident side effect, even at the highest dose, following the biochemical markers evaluated in this study. Noteworthy such significant parasite load reduction was achieved in a short-term treatment (five days).

Altogether, these data, in contrast to the long-term (15 days) intraperitoneal administered **Lapdesf14e**, which decreased by 50% the parasite load in *L. infantum*-infected hamsters [14], revealed the success of the adopted strategy of hit optimization herein adopted and pinpoints **4f** as a new oral antileishmanial compound ready to further advance to the next steps towards the clinical phase of evaluation.

## Supporting information

**S1 Appendix. Additional $^1$H and $^{13}$C NMR information.**
(DOCX)

**S1 Table. Theoretical ADME properties of furoxan and benzofuroxan derivatives.**
(DOCX)

**S1 Fig. Clone pPICZαA_CPB2.8 #2 sequence.** The clone was sequenced using *α factor* (`TAC-TATTGCCAGCATTGCTGC`) and reverse 3'AOX1(`GCAAATGGCATTCTGACATCC`), and the elements inserted on plasmid (EcoRI—blue and Sal I–grey, sites) were identified upstream and downstream of CPB2.8 gene, respectively.
(TIF)

**S2 Fig. Evaluation of biochemical toxicity markers for healthy mice, infected and untreated with *L. infantum* and treated with 4f or miltefosine.** Levels of (A) AST—aspartate aminotransferase; (B) ALP—alkaline phosphatase levels; (C) ALT—alanine aminotransferase; (D) Urea; (E) Total bilirubin levels and (F) Creatinine. The data are expressed as average ± SEM. #: Statistically significant compared to the Infected and Untreated animals ($p < 0.05$). *: Statistically significant compared to the healthy animals ($p < 0.05$). †: Statistically significant compared to the animals treated with the reference drug miltefosine ($p < 0.05$). No statistical significance was observed for creatinine compared to healthy and Untreated animals.
(TIF)

**S1 Raw images.**
(PDF)

## Acknowledgments

The authors are grateful to Dr˚ André Tempone and Dr˚ Felipe Torres for *Leishmania infantum* (MHOM/MA67IMA263) donation.

## Author Contributions

**Conceptualization:** Flávio Henrique-Silva, Jean Leandro dos Santos, Marcia A. S. Graminha.

**Data curation:** Leandro da Costa Clementino, Guilherme Felipe Santos Fernandes, Igor Muccilo Prokopczyk, Wilquer Castro Laurindo, Danyelle Toyama, Bruno Pereira Motta, Amanda Martins Baviera.

**Formal analysis:** Leandro da Costa Clementino, Guilherme Felipe Santos Fernandes, Igor Muccilo Prokopczyk, Danyelle Toyama, Bruno Pereira Motta, Amanda Martins Baviera, Flávio Henrique-Silva, Jean Leandro dos Santos, Marcia A. S. Graminha.

**Funding acquisition:** Marcia A. S. Graminha.

**Investigation:** Wilquer Castro Laurindo.

**Project administration:** Marcia A. S. Graminha.

**Supervision:** Marcia A. S. Graminha.

**Writing – original draft:** Leandro da Costa Clementino, Guilherme Felipe Santos Fernandes, Igor Muccilo Prokopczyk, Flávio Henrique-Silva, Jean Leandro dos Santos, Marcia A. S. Graminha.

**Writing – review & editing:** Leandro da Costa Clementino, Guilherme Felipe Santos Fernandes, Amanda Martins Baviera, Flávio Henrique-Silva, Jean Leandro dos Santos, Marcia A. S. Graminha.

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
