## [Decision Letter · Decision Letter 0]

6 Sep 2021

PONE-D-21-22172Design, Synthesis and Biological Evaluation of N-oxide Derivatives with Potent In Vivo Antileishmanial ActivityPLOS ONE

Dear Dr. Graminha,

Thank you for submitting your manuscript to PLOS ONE. After careful consideration, we feel that it has merit but does not fully meet PLOS ONE’s publication criteria as it currently stands. Therefore, we invite you to submit a revised version of the manuscript that addresses the points raised during the review process.

We look forward to receiving your revised manuscript.

Kind regards,

Bhaskar Saha

Academic Editor

PLOS ONE

“NO”

“This study was supported by Fundação de Amparo à Pesquisa do Estado de São Paulo (FAPESP Process: 2016/09502-7, 2016/06931-4, 2018/17739-2, 2018/11079-0 and 2017/03552-5). Programa de Apoio ao Desenvolvimento Científico da Faculdade de Ciências Farmacêuticas da UNESP – PADC. This study was financed in part by the Coordenação de Aperfeiçoamento de Pessoal de Nível Superior - Brasil (CAPES) and Conselho Nacional de Desenvolvimento Científico e Tecnológico (CNPq). L.C.C. is CNPq PhD student fellow (143334/2017-4). I.M.P is CNPq postdoctoral fellow (153581/2018-2). J.L.S, F.H.S. and M.A.S.G are recipient of a Research Productivity Scholarship from the National Council for Research and Development (CNPq Ref. Process: 304731/2017-0, 311746/2017-9 and 305174/2020-7, respectively). The authors are grateful to Profº André Tempone and Felipe Torres for *Leishmania infantum* (MHOM/MA67IMA263) donation. The funders had no role in study design, data collection and analysis, decision to publish, or preparation of the manuscript.”

“MASG (2017/03552-5 and 2016/06931-4) FAPESP - The São Paulo Research Poundation and (305174/2020-7) CNPq - National Council for Scientific and Technological Development

JLS (2018/11079-0) FAPESP - The São Paulo Research Poundation and (304731/2017-0) CNPq - National Council for Scientific and Technological Development

FHS (311746/2017-9) CNPq - National Council for Scientific and Technological Development

LCC (143334/2017-4) CNPq - National Council for Scientific and Technological Development

IMP (153581/2018-2) CNPq - National Council for Scientific and Technological Development

GFSF (2016/095502-7) (2018/17739-2) - FAPESP - The São Paulo Research Foundation

CNPq: https://www.gov.br/cnpq/pt-br

FAPESP: https://fapesp.br/en/about

Did the sponsors or funders play any role in the study design, data collection and analysis, decision to publish, or preparation of the manuscript?

NO.”

Additional Editor Comments (if provided):

The manuscript reports synthesis of 24 compounds which are tested on Leishmania-infected BALB/c mice for their anti-leishmanial activities. The identified one compound show significant anti-leishmanial activity in both spleen and liver.

However, the manuscript requires several moderations as follows:

1. Please merge Figures 2, 3 and 4 into one - Figure 2

2. Please merge Figures 5 and 6 into one - Figure 3

3. Please merge Figure 7 and 8 into one - Figure 4

4. Please move Figure 9 into supplementary figure.

This will help shorten the manuscript. Please check the composition of the manuscript.

Reviewers' comments:

Reviewer's Responses to Questions

**Comments to the Author**

1. Is the manuscript technically sound, and do the data support the conclusions?

Reviewer #1: Yes

2. Has the statistical analysis been performed appropriately and rigorously? 

Reviewer #1: Yes

3. Have the authors made all data underlying the findings in their manuscript fully available?

Reviewer #1: Yes

4. Is the manuscript presented in an intelligible fashion and written in standard English?

Reviewer #1: Yes

5. Review Comments to the Author

Reviewer #1: Minor Revision

1. Page 24, Line 566, the figure legend needs to be written more elaborately.

2. Figure Number 5 is unreadable.

3. Page no 28, line number 601, figure number 6B and 6C is cited. No figure 6B and 6C is in the paper.

4. Page no 28, line no 607, figure 7 is cited, the description does not match with figure7.

6. PLOS authors have the option to publish the peer review history of their article (what does this mean?). If published, this will include your full peer review and any attached files.

Reviewer #1: No

---

## [Author Response · Author response to Decision Letter 0]

10 Sep 2021

Thank you for all the critiques/suggestions. All answers required are in the letter (Response to Reviewers) that responds each point raised by the academic editor and reviewer. We also performed a careful revision of our Financial Support that should be used to change the online submission form on our behalf.

We hope that with all the additions and modifications made to the text, this version of the manuscript is clearer to the readers and suitable to be now accepted to be published at PLos One.

We are looking forward to hearing from you soon.

---

## [Editor Report · Decision Letter 1]

11 Oct 2021

Design, synthesis and biological evaluation of N-oxide derivatives with potent in vivo antileishmanial activity

PONE-D-21-22172R1

Dear Dr. Graminha,

We’re pleased to inform you that your manuscript has been judged scientifically suitable for publication and will be formally accepted for publication once it meets all outstanding technical requirements.

Kind regards,

Bhaskar Saha

Academic Editor

PLOS ONE
---

## [Editor Report · Acceptance letter]

21 Oct 2021

PONE-D-21-22172R1 

Design, synthesis and biological evaluation of *N*-oxide derivatives with potent *in vivo* antileishmanial activity 

Dear Dr. Graminha:

I'm pleased to inform you that your manuscript has been deemed suitable for publication in PLOS ONE. Congratulations! Your manuscript is now with our production department. 

Kind regards, 

on behalf of

Dr. Bhaskar Saha 

Academic Editor

PLOS ONE